# Postoperative Lifestyle of Patients with Liver Cancer: An Exploratory Study in a Single Center in Taiwan

**DOI:** 10.3390/ijerph19169883

**Published:** 2022-08-11

**Authors:** Sheng-Lei Yan, Chia-Jung Hu, Li-Yun Tsai, Chen-Yuan Hsu

**Affiliations:** 1College of Biotechnology & Bio-Resources, Dayeh University, Changhua 51591, Taiwan; 2Division of Gastroenterology and Hepatology, Department of Internal Medicine, Chang Bing Show Chwan Memorial Hospital, Changhua 51591, Taiwan; 3Department of Nursing, Dayeh University, Changhua 51591, Taiwan; 4College of Nursing, Central Taiwan University of Science and Technology, Taichung 406053, Taiwan

**Keywords:** lifestyle, liver cancer, quality of life, postoperative

## Abstract

Various treatments are available for patients with liver cancer; however, complications after treatment affect their quality of life (QOL). To improve the QOL of patients with liver cancer, this study investigated the postoperative lifestyle of sixty patients at the Liouying District Hospital, Taiwan. A self-reported structured questionnaire and a modified Chinese version of the Health Enhancement Lifestyle Profile (HELP-C) were used to collect the demographic data and to assess patients’ overall postoperative lifestyle, respectively. Significant differences were observed between the overall postoperative lifestyle and the demographic variables of age, ethnicity, education level, marital status, chronic diseases, and postoperative complications (*p* < 0.05). Significant differences (*p* < 0.05) were observed in the HELP-C domains of diet, leisure, and activities of daily living (ADL) between the sexes. The scores for diet (9.66 ± 4.21) and leisure (4.33 ± 2.03) in women were significantly lower (*p* < 0.05) than those in men (13.13 ± 4.98 and 6.17 ± 2.37, respectively), indicating that women with liver cancer should be more concerned about diet and leisure after surgery. However, the score for ADL was significantly higher (*p* < 0.05) in women (17.90 ± 5.15) than in men (13.48 ± 2.56), indicating that men should focus on improving ADL. This research provides reference clinical data on the postoperative lifestyle of patients with liver cancer to improve their QOL.

## 1. Introduction

According to a 2020 report of the Ministry of Health and Welfare, Taiwan, liver cancer has the second-highest mortality rate of all cancers, accounting for 17.5% of all cancer deaths. In terms of sex, the incidence of liver cancer in men is twice that observed in women. Moreover, it is the leading cause of cancer death in men and the second leading cause in women [1]. Due to advancements in medical technology, various treatment options are available for patients with liver cancer. However, after treatment, patients often experience less serious symptoms, such as pain, weight loss, fatigue, or nausea [2,3,4,5,6], to more serious symptoms, such as stomach discomfort and fever, loss of appetite, anxiety, or liver cancer recurrence, thereby affecting their quality of life (QOL) [7,8,9,10]. Studies have reported that patients with liver disease often have unhealthy habits. Therefore, adopting a healthy lifestyle, especially after surgery, is crucial for patients with liver cancer [11,12].

Liver cancer recurrence is common even after patients undergo the most curative surgical resection; early recurrence after liver cancer surgery affects the prognosis of patients [13]. In the past, because of the late diagnosis of liver cancer, the 5-year survival rate was low. Although advancements in medical technology have improved survival rates and reduced postoperative complications and health problems, maintaining a healthy postoperative lifestyle is crucial for patients with liver cancer.

Liver cancer treatments are divided into curative, such as radiofrequency ablation (RFA), and palliative, such as transcatheter arterial embolization (TAE) [14], and these treatments have received much attention in recent years. TAE, the main treatment mode for patients with unresectable liver cancer, can effectively prolong the survival and improve the quality of life (QOL) of such patients [14]. However, clinicians must consider posttreatment complications [14,15,16]. The prognosis for liver cancer depends on multiple factors; therefore, determining the liver cancer treatment that would yield the most favorable prognosis is challenging [16,17,18]. Moreover, further studies must be conducted to explore the effects of postoperative complications and a healthy lifestyle on the QOL of patients with liver cancer.

In 1996, the World Health Organization (WHO) defined QOL as individuals’ degree of satisfaction with their life in the context of the culture and value systems in which they live. These feelings are related to an individual’s goals, expectations, and living standards, and are assessed on the basis of six domains: physical health, mental state, the degree of independence, social relationships, personal beliefs, and the environment [19]. Studies have indicated that the health status of patients with cancer affects their QOL. Along with the treatment process, most patients with liver cancer endure additional postoperative complications and subsequent health problems. Patients’ discomfort affects their QOL because it exerts effects not only at the physical level, but also at the psychological, emotional, spiritual, and environmental levels [19,20,21,22]. This study examined the postoperative lifestyle of patients with liver cancer to explore and improve their QOL.

## 2. Materials and Methods

### 2.1. Study Design and Setting

A questionnaire-based survey was administered to 60 patients with liver cancer aged above 20 years at the Chi Mei Medical Center (Liouying District, Tainan City, Taiwan) from April 2017 to April 2018. The sample size had a power of 0.80 with an alpha significance of 0.05 and an effect size of 0.80 [23]. The study participants were required to speak Mandarin or Taiwanese Hoklo dialect and to have no cognitive disorders that could affect their participation. The study was conducted in accordance with the STROBE (EQUATOR) guidelines. All participants provided written informed consent.

The flowchart of participant recruitment is provided in Figure 1. We assessed 62 participants for eligibility, and two participants were excluded: one declined to complete the survey and one did not meet the language criterion. Therefore, 60 participants were included in this study.

### 2.2. Ethical Considerations

The study was conducted in accordance with the principles of the Declaration of Helsinki and approved by the Institutional Review Board of Chi Mei Medical Center, Tainan, Taiwan (IRB Serial No: 10601-L07).

### 2.3. Measurements

A self-reported structured questionnaire [14] was used to collect demographic data regarding age, sex, ethnicity, education level, marital status, chronic diseases, drug allergy, main treatment, and postoperative complications.

A modified Chinese version of the Health Enhancement Lifestyle Profile (HELP-C) was used to assess participants’ overall postoperative lifestyle. This self-reported structured questionnaire contained 63 questions in seven domains: exercise, diet, social and productive activities, leisure, the activities of daily living (ADL), stress management and spiritual participation, and other health promotion and risk behaviors [14].

Answers to the question “Please tick according to the actual situation in the past three months” were recorded on a 6-point scale (0 = never; 1 = 1 to 2 days per month; 2 = 1 to 2 days per week; 3 = 3 to 4 days per week; 4 = 5 to 6 days per week; and 5 = 7 days per week) [14]. A higher score indicates a healthy lifestyle. The Cronbach’s α for the questionnaire was 0.71, revealing sufficient internal consistency [14].

Exercise was assessed on the basis of nine questions: how many days per week the participant walked for 20 min; performed yoga or a stretching exercise; went to the gym or exercised at home; performed strengthening exercises; biked, jogged, or hiked; performed swimming or surfing; played sports; performed martial art (qigong); and danced [14].

Diet was assessed on the basis of nine questions: how many days per week the participant consumed three servings of healthy foods rich in protein; two servings of healthy foods rich in calcium; three servings of fruits or vegetables; three servings of whole grain foods; two servings of foods high in cholesterol; two servings of foods high in sodium; two servings of foods high in saturated or trans fats; two servings of sweets or desserts; and more than 1500 mL of water [14].

Social and productive activities were assessed on the basis of twelve questions: how many days per week the participant went out with friends or relatives; performed volunteer work; participated in a special activity or hobby group; went to a senior citizen center; participated in a social, cultural, or support group; participated in a political or community activity; participated in an informal or nonacademic class; attended a formal or academic class; connected with family members living apart; participated in family gatherings; took care of grandchildren or other family members; and performed paid work [14].

Leisure was assessed on the basis of nine questions: how many days per week the participant read a newspaper or magazine; watched a favorite TV program; went out to watch sports or movies; engaged in gardening; played chess, bridge, cards, or bingo; wrote diaries, journals, or vignettes; participated in a picnic, fishing, or sailing; engaged in carpentry or automobile or home repairs; and performed cooking as a hobby [14].

The ADL was assessed on the basis of eight questions: how many days per week the participant performed hygiene practices; performed bathing; stayed up late at night; went for food or merchandise shopping; missed a meal; felt that they obtained insufficient rest; performed or helped with housework; and prepared or planned a meal [14].

Stress management and spiritual participation were assessed on the basis of eight questions: how many days per week the participant had a sense of satisfaction in life; performed mood-enhancing activities; spoke to a special person; prayed, worshipped, or chanted; watched spiritual or religious programs; visited a church, temple, or mosque; read spiritual or religious books and meditated; and performed yoga or relaxed [14].

Other health promotion and risk behaviors were assessed on the basis of eight questions: how many days per month the participant consumed three servings of alcohol; smoked five cigarettes in one day; took pain medicine; took over-the-counter drugs; read health-related articles; watched health-related programs; monitored health at home; and attended health promotion programs [14].

### 2.4. Data Analyses

All the statistical analyses were performed using SPSS 22.0 (SPSS Inc., Chicago, IL, USA). The data were analyzed using frequencies, percentages, and an analysis of variance. The resulting content validity index was 0.85, confirming the questionnaire’s appropriateness and applicability. The Cronbach’s α ranged from 0.70 to 0.71, indicating that the questionnaire had sufficient internal consistency. For all statistical tests, significance was set as *p* < 0.05.

## 3. Results

### 3.1. Participant Demographics

The participants, with a mean age of 67.28 years (standard deviation: 10.01 years), were mostly men (*n* = 39; 65%); Hoklo Taiwanese (*n* = 56; 93.3%); college or university graduates (*n* = 28; 46.7%); and married (*n* = 50; 83.3%). Most of them had a diagnosis of chronic diseases (*n* = 39; 65%) and were without a drug allergy (*n* = 53; 88.3%). Regarding treatment modalities, most of them underwent RFA (*n* = 30; 50%) or TAE (*n* = 30; 50%), and postoperative complications were reported in approximately 33.3% of patients (*n* = 20; Table 1).

### 3.2. Demography of Participants and HELP-C Scores

Significant differences were observed between the total HELP-C scores (indicating overall postoperative lifestyle) of the participants and the demographic variables of age (F = 3.39; *p* = 0.00); ethnicity (F = 8.84; *p* = 0.00); education level (F = 9.84; *p* = 0.00); marital status (F = 12.77; *p* = 0.00); chronic diseases (F = 14.57; *p* = 0.00); and postoperative complications (F = 16.77; *p* = 0.00) (Table 2). However, no significant differences were observed between the total HELP-C scores of the participants and the demographic variables of sex, drug allergy, and main treatment (Table 2).

### 3.3. Lifestyle Differences between Sexes

The HELP-C scores for the diet, leisure, and ADL domains differed significantly (*p* < 0.05) between men and women. The scores for diet (9.66 ± 4.21) and leisure (4.33 ± 2.03) in women after liver cancer surgery were significantly lower (*p* = 0.00) than those in men (13.13 ± 4.98 and 6.17 ± 2.37, respectively), indicating that women should be more concerned about diet and leisure after surgery. However, the scores for ADL were significantly higher (*p* = 0.00) in women (17.90 ± 5.15) than in men (13.48 ± 2.56), indicating that men should focus on improving ADL (Table 3; Figure 2). The other categories, namely exercise, social and productive activities, stress management and spiritual participation, and other health promotion and risk behaviors, were not significantly different between the sexes (Table 3).

## 4. Discussion

This study investigated the postoperative lifestyle of patients with liver cancer. Significant differences were observed between the overall postoperative lifestyle of the participants and the demographic variables of age, ethnicity, education level, marital status, chronic diseases, and postoperative complications. This finding is consistent with those of previous studies. Previous studies have reported that patients with liver cancer should aim for a healthy lifestyle after surgery [11,12].

A healthy postoperative lifestyle increases the QOL of patients with liver cancer. Several studies have revealed the significant effect of postoperative complications on the QOL of patients with liver cancer [7,8,9,10,13], a finding that is consistent with that of this study. However, further research is warranted to elucidate the mechanism behind this effect.

Treatment modalities (e.g., RFA and TAE) had no significant effect on the postoperative lifestyle of patients with liver cancer; this result is consistent with those of related studies [16,17,18]. However, further research is required to determine the effect of RFA or TAE on the postoperative complications and lifestyle of patients with liver cancer.

The HELP-C scores for the diet, leisure, and ADL domains were significantly different between men and women, indicating that women with liver cancer should be more concerned about diet after surgery. A previous study reported that women are more likely to be malnourished; therefore, they should be particularly mindful of their overall nutritional status [24]. Another study reported that activities related to leisure and daily living should be promoted in women; this finding is also consistent with that of this study [25]. Men should focus on improving ADL to enhance their well-being, a finding that is consistent with that of a previous study [26]. Therefore, this research identifies several factors (e.g., diet, leisure and ADL) affecting the postoperative QOL of patients with liver cancer; these factors increase patient discomfort by not only affecting physical health but also affecting mental, emotional, spiritual, and environmental health [19,20,21,22]. The WHO defined QOL, which includes the individual’s feelings about themself in life, also affects their subjective experience about QOL [27]. However, the postoperative lifestyle of patients with liver cancer should be investigated further to clarify the mechanisms that are responsible for these effects.

The principal limitation of this study is that it was conducted at a single center and only on 60 patients, meaning the results cannot be generalized to other populations. For the further research suggestion, this study was conducted 4–5 years after the patients’ information was collected in 2017–2018. It will be a more meaningful study if the information can be tracked and the relationship between the survival situation and lifestyle of different patients can be explored.

## 5. Conclusions

Significant differences were observed between the overall postoperative lifestyle of patients with liver cancer and the demographic variables of age, ethnicity, education, marital status, chronic diseases, and postoperative complications. The HELP-C scores for the diet, leisure, and ADL domains were significantly different between men and women. The HELP-C scores for diet and leisure in women after liver cancer surgery were significantly lower than those in men, indicating that women with liver cancer should be more concerned about diet and leisure after surgery. However, the ADL scores of women were significantly higher than those of men, indicating that men should focus on improving ADL. This research provides reference clinical data on the postoperative lifestyle of patients with liver cancer to improve the QOL of such patients.

## Figures and Tables

**Figure 1 ijerph-19-09883-f001:**
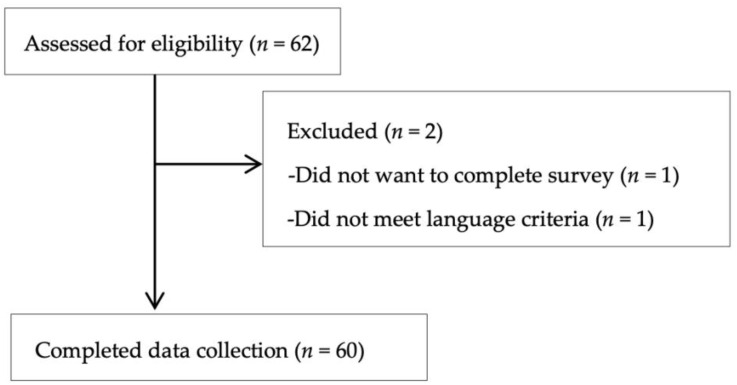
Flowchart of participant recruitment.

**Figure 2 ijerph-19-09883-f002:**
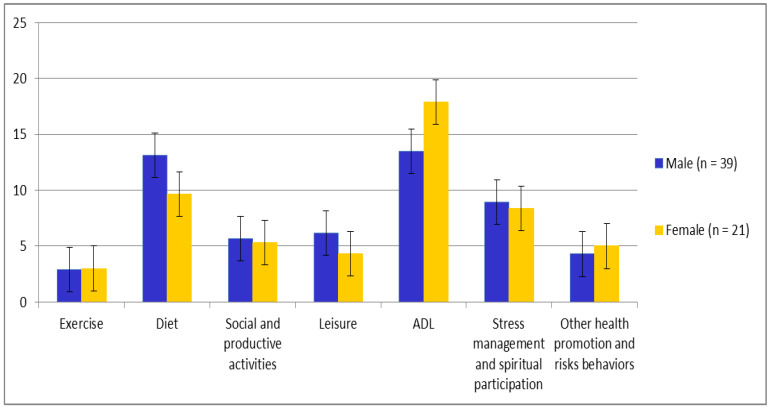
Lifestyle differences between sexes, male (*n* = 39) and female and (*n* = 21).

**Table 1 ijerph-19-09883-t001:** Participant demographics (N = 60).

Variable		*n*	%	
Age				Mean age (SD): 67.28 (10.01)
Gender	Male	39	65.0	
	Female	21	35.0	
Ethnicity	Hoklo Taiwanese	56	93.3	
	Chinese	4	6.7	
Education	None	1	1.7	
	Primary school	6	10	
	Junior high school	8	13.3	
	High school	15	25	
	College/university	28	46.7	
	Research institute	2	3.3	
Marriage	Married	50	83.3	
	Single	1	1.7	
	Widowed	9	15	
Chronic diseases	Yes	39	65	
	No	21	35	
Drug allergy	Yes	7	11.7	
	No	53	88.3	
Main treatment	RFA	30	50	
	TAE	30	50	
Postoperative complications	No	40	66.7	
	Yes	20	33.3	

SD = standard deviation.

**Table 2 ijerph-19-09883-t002:** Demography of participants and HELP-C scores (N = 60).

Items	HELP-C		
	Mean ± SD	F	*p*-Value
Age	-	3.39	0.00 *
Gender		0.06	0.79
Male	54.52 ± 12.23		
Female	53.61 ± 14.45		
Ethnicity		8.84	0.00 *
Hoklo Taiwanese	55.47 ± 12.38		
Chinese	36.75 ± 6.89		
Education		9.84	0.00 *
None	37.00 ± -		
Primary school	40.50 ± 8.91		
Junior high school	38.87 ± 6.55		
High school	56.33 ± 13.27		
College/university	60.33 ± 7.88		
Research institute	66.50 ± 16.26		
Marriage		12.77	0.00 *
Married	56.59 ± 11.34		
Single	78.00 ± -		
Widowed	38.55 ± 12.95		
Chronic diseases		14.57	0.00 *
Yes	49.89 ± 12.89		
No	62.00 ± 8.93		
Drug allergy		0.07	0.79
Yes	55.42 ± 12.47		
No	54.03 ± 13.12		
Main treatment		0.00	0.99
RFA	54.20 ± 12.67		
TAE	54.20 ± 13.43		
Postoperative complications		16.77	0.00 *
No	58.58 ± 11.12		
Yes	45.65 ± 12.17		

* Statistically significant at *p* < 0.05.

**Table 3 ijerph-19-09883-t003:** Lifestyle differences between the sexes (N = 60).

Items	Male	Female		
	Mean ± SD	Mean ± SD	F	*p*-Value
Exercise	2.89 ± 2.50	3.00 ± 3.53	0.01	0.89
Diet	13.13 ± 4.98	9.66 ± 4.21	7.27	0.00 *
Social and productive activities	5.69 ± 3.01	5.33 ± 2.97	0.19	0.66
Leisure	6.17 ± 2.37	4.33 ± 2.03	9.10	0.00 *
ADL	13.48 ± 2.56	17.90 ± 5.15	19.76	0.00 *
Stress management and spiritual participation	8.94 ± 3.84	8.38 ± 4.08	0.28	0.59
Other health promotion and risks behaviors	4.30 ± 2.42	5.00 ± 1.87	1.28	0.26

ADL = activities of daily living. * Statistically significant at *p* < 0.05.

## Data Availability

The raw data supporting the conclusions of this article will be made available by the corresponding author.

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
