# Peer review of "Postoperative Lifestyle of Patients with Liver Cancer: An Exploratory Study in a Single Center in Taiwan"

_ijerph, 2022, doi:10.3390/ijerph19169883_

Round 1

Reviewer 1 Report

The authors conduct an interesting study about the postoperative lifestyle of patients with liver cancer, that will help doctors in real-life practice. 

Minor comment:

The fact that the cohort is based only on 60 patients, should be added as a limitation. 

Author Response

Response 1:

Thanks for your comments.

The revision has revised on the new manuscript, as 4. Discussion, Line 216-217.

Revised as: “The principal limitation of this study is that it was conducted at a single-center and only on 60 patients meaning the results cannot be generalized to other populations.”

Reviewer 2 Report

A well written Exploratory study. Authors focused on one centre to obtain participants and conduct the study which is a limitation to the paper and authors mentioned this point in the discussion. The presentation of the data is not well presented. I would have liked to see some correlation plots for mean differences to show the reader the significance. Please remove the lines in the background of your bar graph. 

The title should be changed to reflect that the study was conducted in a single centre in Taiwan. Discussion, Line 211 beginning "Therefore, this research identifies several factors affecting the postoperative QOL.." needs to mention the factors  Discussion doesn't go into details on how the factors identified can actually reduce QOL. 

Grammar: Please revise sentence in line 67 beginning " Discomfort affects patients’ QOL because it exerts effects this only at the physical level..."

Author Response

Point 1: A well written Exploratory study. Authors focused on one centre to obtain participants and conduct the study which is a limitation to the paper and authors mentioned this point in the discussion. The presentation of the data is not well presented. I would have liked to see some correlation plots for mean differences to show the reader the significance. Please remove the lines in the background of your bar graph.

Response 1:

Thanks for your comments.

The revision has revised on the new manuscript, as Figure 2. Lifestyle differences between sexes, male (n=39) and female and (n=21), Line182-184.

Revised as: remove the lines in the background of the bar graph.

Point 2:The title should be changed to reflect that the study was conducted in a single centre in Taiwan. Discussion, Line 211 beginning "Therefore, this research identifies several factors affecting the postoperative QOL" needs to mention the factors  Discussion doesn't go into details on how the factors identified can actually reduce QOL.

Response 2:

Thanks for your comments.

-The revision has revised on the new manuscript, as title” Postoperative Lifestyle of Patients with Liver Cancer: An Ex-ploratory Study in a single-center in Taiwan.”

-The revision has revised on Line 208-213, as”Therefore, this research identifies several factors (e.g. diet, leisure and ADL) affecting the postoperative QOL of patients with liver cancer; these factors increase patient discomfort by not only affecting physical health but also affecting mental, emotional, spiritual, and environmental health [19–22]. WHO defined the QOL, which includes the individuals feel about them in life, also affects their subjective experience about the QOL (27).”

Point 3: Grammar: Please revise sentence in line 67 beginning " Discomfort affects patients’ QOL because it exerts effects this only at the physical level..."

Response 3:

Thanks for your comments.

The revision has revised on Line 66-68, as”Patients’ discomfort affects their QOL because it exerts effects this not only at the physical level but also at the psychological, emotional, spiritual, and environmental levels [19–22].”

Reviewer 3 Report

This study investigated the postoperative lifestyle of sixty patients at the Liouying District Hospital, Taiwan. It shows some significant differences between the overall postoperative lifestyle and the demographic variables of ethnicity, education level, marital status, chronic diseases, and postoperative complications. This study may provide some cues to improve the patients’ QOL. The following are the questions.

1.      Line 99-104, it is stated that the survey items in the questionnaire include health promotion behaviors and risk behaviors. However, in line 102 at the end of this paragraph, it is pointed out that a higher score represents a healthy lifestyle, which is ambiguous. Please explain specific implementation of the score evaluation. Did you do the plus or minus points? A clear method of how to evaluate the points should be described.   

2.      Results section 3.3, lines 172-183, two conclusions on gender derived from the evaluation scores are inconsistent. In terms of eating and leisure, the low score of women means that women pay more attention to it?  In ADL, low scores of men indicate that men do not pay enough attention to each other?

3.      Table 2, 67.28 is not the HELP-C score for age. It should not be included in this Table. And I think as age is quite an important factor that influences the lifestyle of patients, HELP-C score for age should be included.

4.      It has been 4-5 years since the patients information was collected in 2017-2018 years. It will be a more meaningful study if the information can be tracked and the relationship between the survival situation and lifestyle of different patients can be explored.

Author Response

Point 1:

  1. Line 99-104, it is stated that the survey items in the questionnaire include health promotion behaviors and risk behaviors. However, in line 102 at the end of this paragraph, it is pointed out that a higher score represents a healthy lifestyle, which is ambiguous. Please explain specific implementation of the score evaluation. Did you do the plus or minus points? A clear method of how to evaluate the points should be described.

Response 1:

Thanks for your comments.

The revision has revised on the new manuscript, as, Line 98-102, as” Answers to the question “Please tick according to the actual situation in the past three months” were recorded on a 6-point scale (0 = never; 1 = 1 to 2 days per month; 2 = 1 to 2 days per week; 3 = 3 to 4 days per week; 4 = 5 to 6 days per week; and 5 = 7 days per week) [14]. A higher score indicates a healthy lifestyle. The Cronbach’s α for the ques-tionnaire was 0.71, revealing sufficient internal consistency [14].”

Point 2: Results section 3.3, lines 172-183, two conclusions on gender derived from the evaluation scores are inconsistent. In terms of eating and leisure, the low score of women means that women pay more attention to it?  In ADL, low scores of men indicate that men do not pay enough attention to each other?

Response 2:

Thanks for your comments.

-The revision has revised on Section 3.3, Line 170-179, as” The HELP-C scores for the diet, leisure, and ADL domains differed significantly (p < 0.05) between men and women. The scores for diet (9.66 ± 4.21) and leisure (4.33 ± 2.03) in women after liver cancer surgery were significantly lower (p = 0.00) than those in men (13.13 ± 4.98 and 6.17 ± 2.37, respectively), indicating that women should need more concerned about diet and leisure after surgery. However, the scores for ADL was significantly higher (p = 0.00) in women (17.90 ± 5.15) than in men (13.48 ± 2.56), indicating that men should focus on improving ADL (Table 3; Figure 2). The other categories, namely exercise, social and productive activities, stress management and spiritual participation, and other health promotion and risk behaviors, were not significantly different between the sexes (Table 3).”

Point 3: Table 2, 67.28 is not the HELP-C score for age. It should not be included in this Table. And I think as age is quite an important factor that influences the lifestyle of patients, HELP-C score for age should be included.

Response 3:

Thanks for your comments.

-The revision has revised on Line 167-168,as “Table 2. Demography of participants and HELP-C scores (N = 60).”

Point 4: It has been 4-5 years since the patients information was collected in 2017-2018 years. It will be a more meaningful study if the information can be tracked and the relationship between the survival situation and lifestyle of different patients can be explored.

Response 4:

Thanks for your comments.

-The revision has revised on Line 216-221,as” The principal limitation of this study is that it was conducted at a single-center and only on 60 patients meaning the results cannot be generalized to other populations. For the further research suggestion, this study has been 4-5 years since the patients information was collected in 2017-2018 years. It will be a more meaningful study if the information can be tracked and the relationship between the survival situation and lifestyle of different patients can be explored.”

Reviewer 4 Report

The analysis is interesting

Please clarify this concept: (Line 74: all participants had undergone liver resection surgery ..... Line 156-157: Regarding treatment modalities, most of them underwent RFA (n = 30; 50%) or TAE (n = 30; 50%),  and postoperative complications were reported in approximately 33.3% of patients) Did the patients underwent liver resection followed by RFA or TAE?

Did you analyze cirrhotic status and ECOG Performance status of the patients? 

Author Response

Point 1: Please clarify this concept: (Line 74: all participants had undergone liver resection surgery ..... Line 156-157: Regarding treatment modalities, most of them underwent RFA (n = 30; 50%) or TAE (n = 30; 50%),  and postoperative complications were reported in approximately 33.3% of patients) Did the patients underwent liver resection followed by RFA or TAE? Did you analyze cirrhotic status and ECOG Performance status of the patients?

Response 1:

Thanks for your comments.

The treatment modalities that is RFA or TAE. We are not analyze for those cirrhotic status and ECOG Performance status of the patients.

-The revision has revised on the new manuscript, as, Line 72-75, as ”A questionnaire-based survey was administered to 60 patients with liver cancer aged above 20 years at the Chi Mei Medical Center (Liouying District, Tainan City, Taiwan) from July 2017 to April 2018. The sample size had a power of 0.80 with an alpha significance of 0.05 and an effect size of 0.80 [23].”

Round 2

Reviewer 3 Report

Accept